# Novel Eco-Friendly Tannic Acid-Enriched Hydrogels-Preparation and Characterization for Biomedical Application

**DOI:** 10.3390/ma13204572

**Published:** 2020-10-14

**Authors:** Beata Kaczmarek, Oliwia Miłek, Marta Michalska-Sionkowska, Lidia Zasada, Marta Twardowska, Oliwia Warżyńska, Konrad Kleszczyński, Anna Maria Osyczka

**Affiliations:** 1Department of Biomaterials and Cosmetics Chemistry, Faculty of Chemistry, Nicolaus Copernicus University, 87-100 Toruń, Poland; 296559@stud.umk.pl (L.Z.); 296558@stud.umk.pl (M.T.); 2Department of Biology and Cell Imaging, Institute of Zoology and Biomedical Research, Faculty of Biology, Jagiellonian University in Kraków, 30-387 Kraków, Poland; oliwia.milek@student.uj.edu.pl (O.M.); anna.osyczka@uj.edu.pl (A.M.O.); 3Department of Environmental Microbiology and Biotechnology, Faculty of Biology and Environmental Protection, Nicolaus Copernicus University in Toruń, 87-100 Toruń, Poland; mms@umk.pl (M.M.-S.); oliwia.warzynska@vp.pl (O.W.); 4Department of Dermatology, University of Münster, Von-Esmarch-Str. 58, 48149 Münster, Germany; konradk1977@gmail.com

**Keywords:** tannic acid, sodium alginate, hydrogel, wound dressings

## Abstract

Sodium alginate and tannic acid are natural compounds that can be mixed with each other. In this study, we propose novel eco-friendly hydrogels for biomedical applications. Thus, we conducted the following assessments including (i) observation of the structure of hydrogels by scanning electron microscope; (ii) bioerosion and the concentration of released tannic acid from subjected material; (iii) dehydrogenase activity assay to determine antibacterial activity of prepared hydrogels; and (iv) blood and cell compatibility. The results showed that hydrogels based on sodium alginate/tannic acid exert a porous structure. The immersion in simulated body fluid (SBF) results in the biomineralization process occurring on their surface while the bioerosion studies revealed that the addition of tannic acid improves hydrogels’ stability proportional to its concentration. Besides, tannic acid release concentration depends on the type of hydrogels and the highest amount was noticed for those based on sodium alginate with the content of 30% tannic acid. Antibacterial activity of hydrogels was proven for both Gram-negative and Gram-positive bacteria, the hemolysis rate was below 5% and the viability of the cells was elevated with an increasing amount of tannic acid in hydrogels. Collectively, we assume that obtained materials make the imperative to consider them for biomedical applications.

## 1. Introduction

Polymers may be proposed as raw compounds for the preparation of the hydrogels. Thus, hydrogels are a physiochemical form of materials that are characterized by a high ability to swell. They may absorb water as well as other hydrophilic liquids, also with dissolved natural compounds. The moist hydrogels assist in protecting the body against infections and promoted wound healing processes [1,2,3]. By hydrogels, the exchange process of water and oxygen is able to do what is essential in cases of skin defects treatments [4]. The exchange is possible because of the presence of pores in the hydrogel structure as well as in the presence of hydrophilic groups. Besides, it is possible to incorporate active compounds within hydrogels and from where they may be released from the defected site. Thus, there is a need to search for bioactive hydrogels that may be proposed as wound dressings. They should be biocompatible without the cytotoxicity effect for surrounded cells [5] as well as optimal tissue integration [6]. For such reasons, hydrogels based on natural polymers are mainly used in scientific research to be tested for wound healing.

Sodium alginate consists of β-(1-4) linked d-mannuronic acid and α-(1-4) linked *L*-guluronic acid. It is an anionic polysaccharide that has been proposed for biomedical application due to its high biocompatibility and low toxic profile [7,8,9]. The negatively charged groups present in the polymeric chain are easy to be cross-linked by the cation. Sodium alginate-based hydrogels may be formed by ionotropic gelation with divalent ions such as Ca^2+^, Cu^2+^, Pb^2+^, Cd^2+^, Ba^2+^ [10,11,12], trivalent Ce^3+^, Al^3+^, Fe^3+^ ions [13,14,15] as well as quadrivalent ones including Zr^4+^ [16,17]. The type of cross-linking agent used for hydrogels affects the physicochemical properties of the final material as well as their biological features. Namely, hydrogel preparation by calcium ions is the most common method, as sodium ions present in sodium alginate salt may be easily exchanged by Ca^2+^. Each calcium ion attaches to two polymer strands. As ionic interactions are formed, sodium alginate is cross-linked and thereby the improvement of material properties is observed. For instance, the addition of phenolic acid to polymeric matrixes has been proposed as a cross-linking method and as the factors improving their new properties. Tannic acid or phenolic acid are widely studied, and have been proven as protein [18,19,20,21] and polysaccharides cross-linkers [22,23,24]. Prepared materials possess novel properties such as antimicrobial and antioxidant activity and sodium alginate-based hydrogels enriched by tannic acid are novel. Therefore, in our previous studies, we have tested tannic acid as a chitosan cross-linker, resulting in obtaining stable films by solvent evaporation [25]. Tannic acid/sodium alginate hydrogels have been previously fabricated by cross-linking with Pb^2+^, Hg^2+^, and Cr^4+^ ions [26]. The novelty of the present work is the preparation of hydrogels based on sodium alginate cross-linked with calcium ions and enriched by tannic acid on the other side.

## 2. Results

### 2.1. Scanning Electron Microscope

The structures of sodium alginate/tannic acid hydrogels are shown in Figure 1. Hydrogels have a porous structure with interconnected pores. Besides, hydrogels based on sodium alginate and with the addition of 10% tannic acid showed an increase in pore diameter as the degradation process occurred. After 7 days in simulated body fluids (SBF), small spots on the surface of the materials were observed. As hydrogels were immersed in SBF, the biomineralization process occurred. Furthermore, the changes in morphology were not observed for hydrogels of 80SA/20TA and 70SA/30TA as tannic acid addition improves the material stability. The biomineralization process also occurred on these types of hydrogels.

### 2.2. Bioerosion

The percentage of hydrogels’ weight changes is shown in Figure 2. Weight loss of hydrogels immersed in SBF is the result of the material degradation process. The highest weight loss after 48 h was noticed for hydrogels obtained from raw sodium alginate. The small addition of TA (10%) results in stability improvement. Materials with a higher tannic acid concentration showed a decrease in stability but were better than hydrogels without TA. The results may suggest that higher TA addition did not result in the effective cross-linking of sodium alginate where 10% addition was enough to cross-link all hydrophilic groups in the polymeric chain.

### 2.3. Tannic Acid Release

Tannic acid release in time dependence is shown in Figure 3 where the released TA concentration depends on the initial tannic acid content in the hydrogel. The released concentration increased with increasing immersion time. The maximum was detected after 90 min immersion in phosphate buffered saline (PBS). Until that time, hydrogels swelled and as a result, the TA released concentration was the highest. After 90 min, the concentration decreased as the degradation process was initiated.

### 2.4. Dehydrogenase Activity 

Tetrazolium salts may be used for measuring the metabolic activity of cells. Tetrazolium salt-based assay is used for quantification of viable and metabolic active cells in planktonic cultures as well as in biofilms [27]. In this reaction, tetrazolium compound (3-(4,5-dimethylthiazol-2-yl)-5-(3-carboxymethoxyphenyl)-2-(4-sulfophenyl)-2H-tetrazolium, inner salt; MTS) was reduced to formazan product. Enzymes that catalyze the oxidation of a substrate in respiratory processes are dehydrogenases [28]. The results of the dehydrogenase activity of bacterial suspension after contact with materials are presented in Figure 4. It was found that prepared hydrogels decrease the dehydrogenase activity of both Staphylococcus aureus and Escherichia coli. However, materials with tannic acid were more efficient and more greatly inhibit the activity of these enzymes compared to sodium alginate alone. Moreover, the dehydrogenase activity was lower with the time of contact with hydrogels. Staphylococcus aureus was more prone to hydrogels compared to E. coli. The dehydrogenase activity of S. aureus after 4 h contact with 70SA/30TA was only 22.23% and 33.69% for E. coli. Tannic acid addition increases the effect of sodium alginate on the bacteria electron transport system. 

### 2.5. Blood Compatibility

The hemolysis rate of hydrogels is listed in Table 1. The increase of tannic acid content in hydrogel results in the increasing hemolysis rate. According to the ASTM, F756-00 standard as nonhemolytic material may be called if the hemolysis rate is below 2%. A rate between 2–5% allows classifying the material as slight hemolytic. Materials with a hemolysis rate below 5% may be proposed for biomedical applications. Thereby, all hydrogels obtained by us may potentially be used for the wound dressing fabrication.

### 2.6. Results of Human BMSC, HaCaT, and MNT-1 Growth/Survival on the Studied Materials

The principle of the MTS assay is to evaluate the number of metabolically active cells. The absorbance of the product is directly proportionate to the number of viable cells. The results of the MTS assay for the cells cultured on tested materials are shown in Figure 5. In all cell types (BMSC, HaCaT, and MNT-1), the addition of tannic acid increased the cell viability, although not all results were statistically significant. For bone marrow stromal cells, the addition of 30% tannic acid to the sodium alginate base significantly increased metabolic activity. Hydrogels with the addition of 20% and 30% of tannic acid stimulated a significant increase in the viability of skin-related cell lines: HaCaT and MNT-1; however, the growth was more pronounced for melanoma cells.

Tannic acid added to sodium alginate acts as a cross-linker and thereby improves the physicochemical properties of hydrogels. Moreover, it has bioactive properties. It has been reported by Ninan [29] that agarose/tannic acid hydrogels did not show any cytotoxic effect on 3T3 fibroblast cell lines. Du et al. [30] reported that the hydrogels based on modified sodium alginate and tannic acid had not only shown good cell affinity but also promoted cell adhesion and spreading. The studies of hydrogel particles fabricated from tannic acid on L929 fibroblast and H1299 lung cancer cells confirmed that tannic acid stimulates cell proliferation compared to the control [31]. Hydrogels based on sodium alginate, tannic acid, and chemically cross-linked polyacrylamide were reported [32] to be biocompatible with L929 cells, which displayed a healthy growth after 72 h in culture with the hydrogel extracts. Results obtained by us are relevant to the mentioned studies as tannic acid addition to sodium alginate results in the increase of cell viability, possibly due to the improvement in the mechanical properties of the hydrogels, giving the cells a good base for migration and spreading.

## 3. Discussion

Bacterial infections are one of the most serious problems in material implantation. The increase of antimicrobial resistance has a significant impact on public health, global development, and even the global economy. By 2016, worldwide, around 700,000 deaths per year were the result of infections caused by antimicrobial-resistant microorganisms. During one of the lectures carried out by Jim O’Neal, it is estimated that by 2050 this figure will reach 10 million deaths each year if the problem of antimicrobial resistance will not be resolved. Besides, the constant increase in antimicrobial resistance is the reason for longer hospital stays and expensive intensive care. Furthermore, some of these compounds have antiviral activity, which in the current situation takes on a new, innovative meaning. This causes and will continue to cause a significant financial burden.

Sodium alginate-based hydrogels may be easily prepared by cross-linking with calcium ions. However, such hydrogels do not show antibacterial activity. Sodium alginate-based hydrogels may be also obtained by mixing with other polymers such as, e.g., polyvinyl alcohol [33]. Obtained materials possess good biocompatibility and burn-healing properties. Moreover, such systems of PVA/SA have been enriched by cellulose to improve the physicochemical properties of hydrogels [34]. However, such materials do not protect against bacteria. We propose simple hydrogels based on SA with tannic acid. The addition of tannic acid results in the activity against S. aureus and E. coli, which marks out our hydrogels. Moreover, as they are composed of natural raw materials, they are eco-friendly. Mixing sodium alginate and tannic acid with the further cross-linking process by calcium ion addition is novel and allows for the preparation of hydrogels with a fast and low-cost method. The obtained materials were stable at 37 °C and showed significant compatibility with cells. Furthermore, prepared hydrogels had an antibacterial effect against Gram-positive and Gram-negative bacteria. This results in proposed hydrogels combining to make up important features such as biocompatibility, antibacterial effectiveness, and safety for the environment. The proposed hydrogels would protect wounds against bacterial infections and at the same time would be safe for cells, or could even enhance their proliferation.

Sodium-alginate hydrogels may be enriched by nano-hydroxyapatite and thereby be dedicated to bone regeneration [35]. We have studied the behavior of hydrogels in simulated body fluids to determine if the biomineralization process occurs. As it was observed on the SEM images, the calcium phosphate has been precipitated in the surface of the hydrogels. This indicates that they may be studied for bone regeneration in the future. We truly believe that hydrogels fabricated by us may be promising novel eco-friendly hydrogels for tissue regeneration and in vivo studies are planned for the future.

## 4. Materials and Methods

### 4.1. Reagents

Sodium alginate (SA) and tannic acid (TA) were purchased from Sigma-Aldrich (Sigma-Aldrich, Darmstadt, Germany). They were dissolved separately in 0.1 M acetic acid at the final concentration of 2%. Calcium chloride was purchased from STANLAB (STANLAB, Lublin, Poland).

### 4.2. Hydrogel Preparation

Both polymer solutions (sodium alginate and tannic acid) were mixed in the ratio 90/10, 80/20, and 70/30 on the magnetic stirrer. Next, mixtures were placed in the dialysis tube (MWCO12000–14000) and dialysis was carried out for 1 week against 5% CaCl_2_ solution, which was changed every 3 days. Calcium ions were used for the cross-linking process of obtained hydrogels resulting in obtainment of stable soft materials. Hydrogels with a tannic acid content higher than 30% were not stable. The cross-linking method by using Ca^2+^ ions is effective when the main component of the hydrogel is sodium alginate. Hydrogels without tannic acid were tested as a control.

### 4.3. Scanning Electron Microscope

The structure of the obtained hydrogels was observed by the scanning electron microscope (SEM) (LEO Electron Microscopy Ltd., Cambridge, UK). Hydrogels were dried by lyophilization and placed on the holder. They were covered with gold for observation. SEM images were made before and after the degradation process.

### 4.4. Bioerosion

Hydrogels were immersed in SBF solution (pH = 7.4). Hydrogels were weighed before and after 1, 4, 8, 24, and 48 h immersion. During the experiment, they were stored in an incubator at 37 °C. The percentage weight change in time was calculated. The experiment was repeated three times and the results are presented as mean values.

### 4.5. Tannic Acid Release

The tannic acid release was carried out in three different types of conditions—simulated body fluid (SBF; pH = 7.4). The total content of polyphenols was determined by the Folin-Ciocalteu method [36]. To perform this analysis, 1 mL of Na_2_CO_3_ was mixed with 0.5 mL of Folin-Ciocalteu reagent and 1 mL of sample. Then the mixture was filled with distilled water to complete 10 mL and was stored at 40 °C for 30 min. Then the absorbance of the samples was measured at 725 nm, in triplicate for separately prepared materials with the use of a UV–Vis spectrophotometer (UV-1800, Shimadzu, Reinach, Switzerland).

### 4.6. Dehydrogenase Activity

The microorganisms *Staphylococcus aureus* ATCC6538, *Escherichia coli* ATCC8739 were chosen as model strains. The analysis was performed according to the method described by Walczak et al., 2020 with small modifications [28]. The overnight culture was centrifuged and cell pellets were suspended in saline salt to obtain OD=1 in McFarland scale. Materials were cut into the pieces in roller shape with around 0.5 cm^3^ volume, placed in 24 well plates, and sterilized using a UV light for 20 min [28]. Next, the nutrient broth was sterilized (composition (g/L): yeast extract (3), tryptone (5), and cell culture were added to materials and incubated at 37 °C for 60 and 180 min. The positive control was prepared without materials. Dehydrogenase activity tests were performed using CellTiter 96 AQ One Solution Cell Proliferation Assay (Promega, Walldorf, Germany) according to manufacturer guides. To sterile 96 well plates 20 μL of CellTiter 96 AQ One Solution Reagent was added to 100 μL of medium taken from the well with samples. The plate was incubated at 37 °C for 90 min. After incubation, absorbance was measured at 490 nm on a Microplate reader (Thermo Fisher Scientific, Waltham, MA, USA). It was assumed that the dehydrogenase activity without contact with hydrogels was 100%. Each sample was checked in four repetitions and results were shown as mean and standard deviation.

### 4.7. Blood Compatibility

A blood compatibility test was prepared by using contact methods. Anticoagulated sheep blood (0.2 mL) was added to the physiological saline solution (10 mL) containing different specimens in a roller shape (around 0.5 cm^3^ volume). Positive and negative samples were prepared by adding the same volume of fresh blood to water and physiological saline, respectively. The mixture were incubated at 37 °C and after 1 h the suspension was centrifuged (1000 r.p.m., 10 min). The absorbance of each supernatant was measured by Multiscan FC (Thermo Fisher Scientific, Waltham, MA, USA) at 540 nm. Each sample was tested in triplicate. The hemolysis rate was calculated using the following equation:rate of hemolysis [%] = ([OD]specimen − [OD]negative)/([OD]positive − [OD]negative) × 100%

### 4.8. Cultures of Human Bone Marrow Stromal Cells (BMSC), HaCaT and MNT-1

All cell culture reagents were obtained from Gibco/ThermoFisher Scientific unless stated otherwise. Bone marrow stromal cells (BMSC) used for this study were obtained based on the approval of the Institutional Review Board protocol nr 1072.6120.254.2017. The bone marrow samples were harvested from 42–58-year old patients of both genders, scheduled for primary hip replacement surgery. The mononuclear cell fractions were collected using Ficoll-Paque Premium (GE Healthcare, Sigma-Aldrich, Darmstadt, Germany) as previously described [37] and expanded in medium containing alpha-MEM, 10% FBSQ (Biological Industries) and 1% ZellShield (Minerva Biolabs) at 37 °C, 5% CO_2_, humidified. After expansion, cells at passages 2–3 were used for experiments. HaCaT (human keratinocytes) cell line was expanded in a medium containing alpha-MEM, 5% FBS and 1% ZellShield. MNT-1 (human melanoma) cell line was cultured in DMEM high glucose (Sigma), 20% inactivated FBS, and 1% ZellShield (Minerva Biolabs) with the addition of 1% *L*-glutamine, 1% HEPES, 1% sodium pyruvate, and 1% MEM Non-essential Amino Acid (all reagents from Sigma). The same dedicated media were used while culturing cells on tested materials. Materials were cut into the small pieces as described in the 2.5 Dehydrogenase activity study and were sterilized for 20 min in 75% EtOH, followed by rinsing for 20 min with sterile PBS (BioShop). Sterile materials were transferred to 12-well plates (NEST). All types of cells used in these experiments were rinsed first with PBS and detached using 0.25% Trypsin-EDTA. Cells were seeded at the density of 4 × 10^4^ cells directly onto the material surface in a small volume of the medium. Following 30 min pre-incubation in a culture incubator, 2 mL of medium was added to each well. Media were exchanged on day 2 cultures. The metabolic activity of cells, which is directly proportional to live cell number, was evaluated on day 6 cultures using an MTS assay (CellTiter 96^®^ Aqueous One Solution Cell Proliferation Assay, Promega, Walldorf, Germany) as previously described [38]. Briefly, MTS reagent was diluted 10× in phenol-free MEM and 400 μL aliquots of diluted MTS were added to each well. All tests were performed in triplicates for separately prepared materials. Results were statistically evaluated using one-way ANOVA and post-hoc Tukey with *p* < 0.05.

## 5. Conclusions

Sodium alginate and tannic acid may be mixed with each other to obtain stability by cross-linking with Ca^2+^ ions. Porous hydrogels promote the biomineralization process, which is important when the material is considered for biomedical application. Proposed materials of alginate/tannic acid hydrogels cross-linked by calcium ions have an antibacterial effect which provides a great advantage compared with pure alginate-based materials. The tannic acid released from the material indicates its antibacterial effect against both Gram-negative and Gram-positive bacteria. The criteria in the material consideration for biomedical application is the hemolysis rate below 5%, which was noticed for our hydrogels. Moreover, the increase in cell viability with an increasing amount of tannic acid in hydrogels was observed. It may be assumed that the proposed preparation method leads to obtainment of the eco-friendly and biocompatible materials mainly used as wound dressings.

## Figures and Tables

**Figure 1 materials-13-04572-f001:**
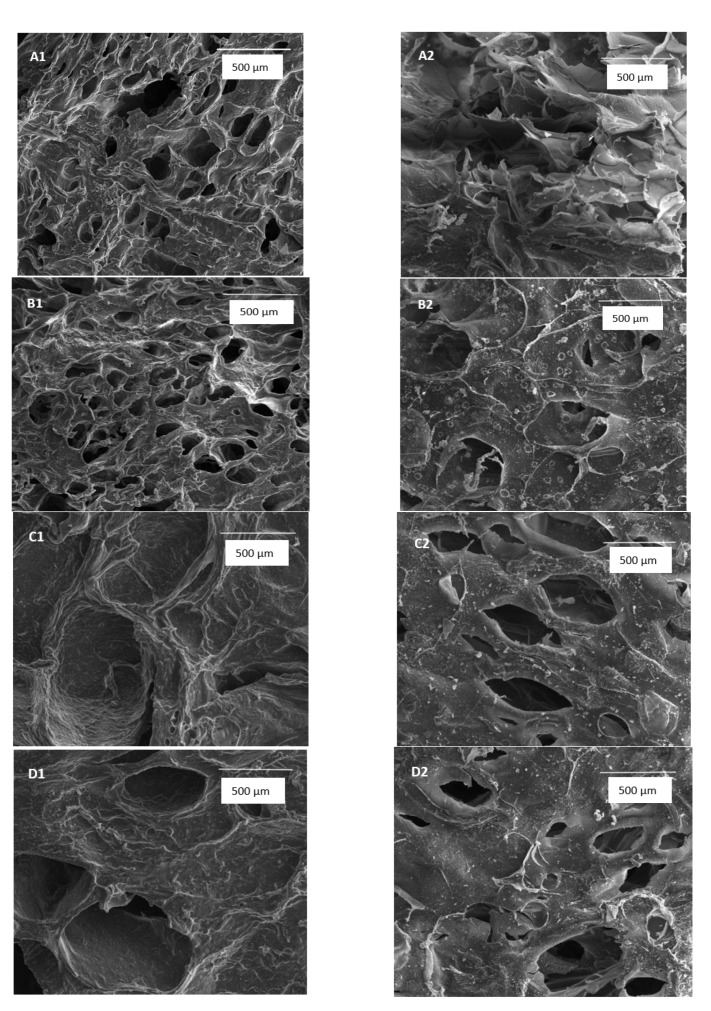
SEM images of (**A**) 100SA (**B**) 90SA/10TA (**C**) 80SA/20TA, and (**D**) 70SA/30TA before immersion (**1**) and after 7 days in SBF solution (**2**) (magnification: 150×).

**Figure 2 materials-13-04572-f002:**
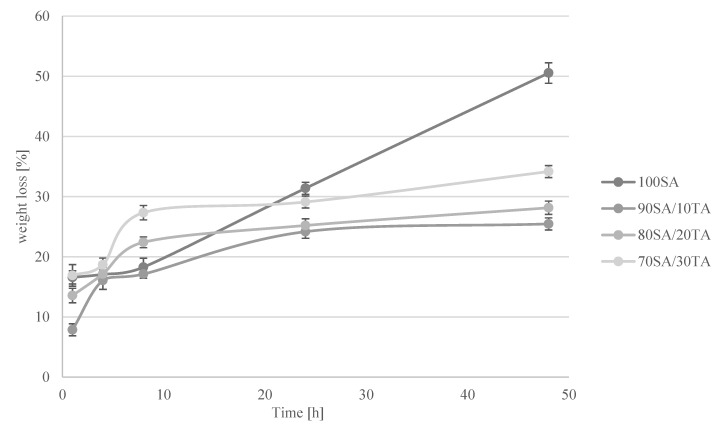
Weight loss [%] of hydrogels immersed in SBF for 1, 4, 8, 24, and 48 h.

**Figure 3 materials-13-04572-f003:**
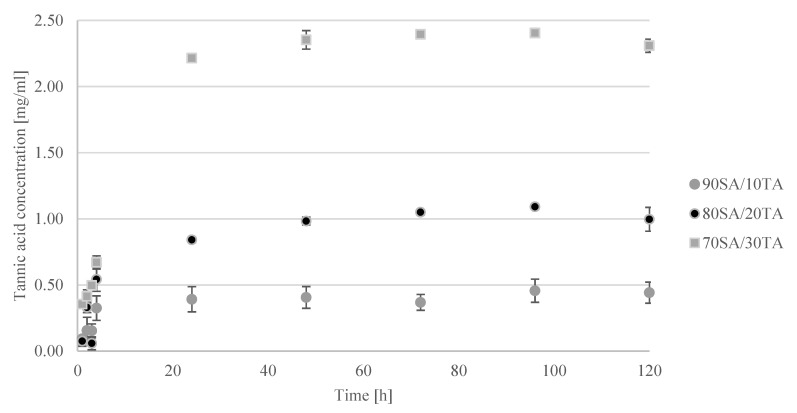
The concentration of tannic acid released from hydrogels when immersed in phosphate buffered saline (PBS).

**Figure 4 materials-13-04572-f004:**
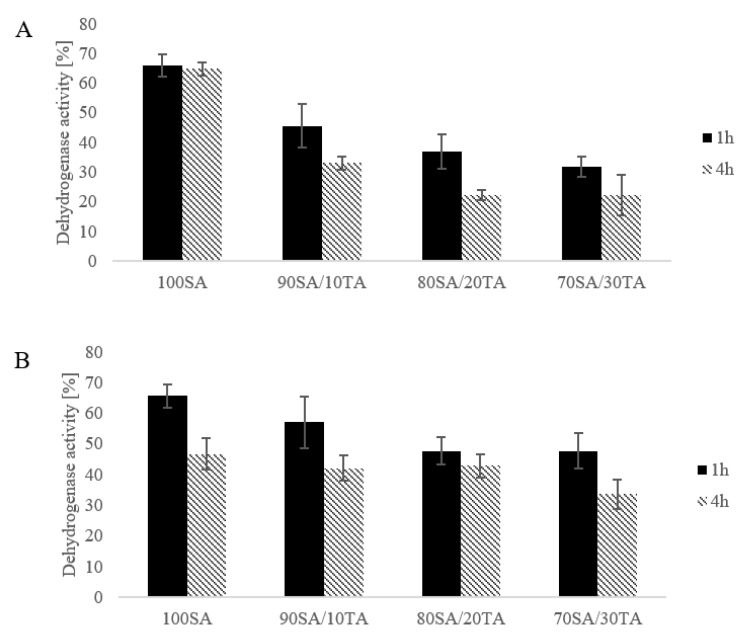
Dehydrogenase activity against Staphylococcus aureus (**A**) and Escherichia coli (**B**) of hydrogels after 1 h and 4 h exposure.

**Figure 5 materials-13-04572-f005:**
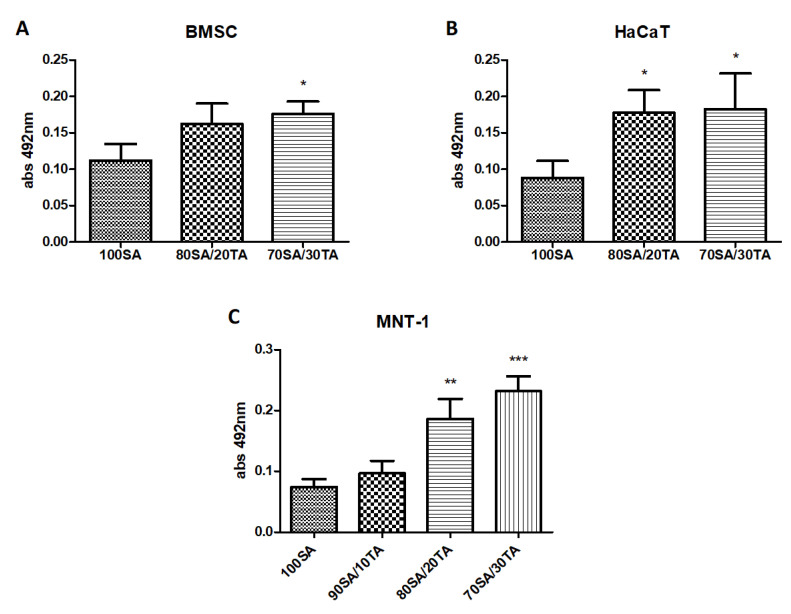
Metabolic activity of human BMSC (**A**), HaCaT (**B**), and MNT-1 (**C**) cells after 6-day culture on the materials composed of sodium alginate (SA) and tannic acid (TA) in ratios 100SA, 90SA/10TA, 80SA/20TA, and 70SA/30TA. Asterisk (** *p* < 0.05; *** *p* < 0.001) indicates statistically significant differences vs. 100SA (control).

**Table 1 materials-13-04572-t001:** The rate of hemolysis for hydrogels based on sodium alginate with and without the addition of tannic acid.

Specimen	Hemolysis Rate (%)
100SA	0.56 ± 0.14
90SA/10TA	0.32 ± 0.20
80SA/20TA	0.92 ± 0.45
70SA/30TA	2.19 ± 1.42

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
