# Peer review of "Novel Eco-Friendly Tannic Acid-Enriched Hydrogels-Preparation and Characterization for Biomedical Application"

_materials, 2020, doi:10.3390/ma13204572_

Round 1

Reviewer 1 Report

Novel eco-friendly tannic acid-enriched hydrogels - preparation and characterization for biomedical application

In their paper, Kaczmarek et al, describe the production and initial testing for potential biomedical applications of a hydrogel composed of Ca2+ alginate and tannic acid.

In my opinion, this is really interesting and a valuable contribution to the field of hydrogel-based materials for biomedical applications. However, I would recommend following points of revisions to improve the manuscript prior to publication.

Major revisions:

The English should be substantially improved; I would strongly recommend asking a native English-speaking colleague to revise the manuscript or take advantage of an English-editing service.

In the introduction, the authors state that the combination of calcium-alginate and tannic acid is novel, however, there are publications describing the combination of alginate-based hydrogels and tannic acid; and the use of calcium instead of sodium. The authors should first mention these previous works in their introduction and also in the discussion and they should describe more clearly, what makes their approach unique.

I have a question concerning the weight loss: why is the starting weight different between the materials and how often was the experiment repeated. Are these mean values?

A related question to the tannic acid release: in materials and methods, the authors state, that triplicate measurements were performed – does that mean, that three independent preparations of the hydrogel combinations were analysed? In my Opinion, this should be the case. The same is true for the biological experiments: at least three independent (independent material preparations, independent cell and bacteria preparations) should be analyzed to account for some biological variations.

For the tests on antibacterial effects, as well as for the blood and cell compatibility the hydrogel materials were sterilized by UV-irradiation, and 75% ethanol treatment, respectively. Does this treatment have effects on (i) structure/integrity of the material (e.g. does the pore size change; is tannic acid released by ethanol treatment etc.), and (ii) on the subsequent test (i.e. was the ethanol removed completely from the hydrogel, before incubating the material with the cells?)

To show an antibacterial activity of the hydrogel (in its envisioned form for example as a wound dressing), I would suggest to perform an agar diffusion test and to show the inhibition zone of the different material compositions, additionally.  

The discussion is very basal and should be considerably extended to comprise discussions on:

                Comparison of the results of this study to results of former studies analyzing similar materials; is the “novel” approach better, is the outcome different

                What to expect from the tannic acid-alginate hydrogel: based on the known activities of tannic acid, what would be potential modes of application.

Author Response

  1. The English should be substantially improved; I would strongly recommend asking a native English-speaking colleague to revise the manuscript or take advantage of an English-editing service.
  • Thank you very much for this valuable hint. Language has been corrected.
  1. In the introduction, the authors state that the combination of calcium-alginate and tannic acid is novel, however, there are publications describing the combination of alginate-based hydrogels and tannic acid; and the use of calcium instead of sodium. The authors should first mention these previous works in their introduction and also in the discussion and they should describe more clearly, what makes their approach unique.
  • Thank you very much for this valuable remark. We agree that the statement “novel” is very strong. Thereby, we reformulated the introduction section accordingly:

…..Tannic acid/sodium alginate hydrogels have been previously fabricated by the cross-linking with Pb(II), Hg(II) and Cr (VI) ions [26]. The novelty of present work is the preparation of hydrogels based on sodium alginate cross-linked by calcium ions and enriched by tannic acid on the other side…..”

[26] Zhang, Y.; He, F.; Xiaoli, Li. Three-dimensional composite hydrogel based on polyamine zirconium oxide, alginate and tannic acid with high performance for Pb(â…ˇ), Hg(â…ˇ) and Cr(â…Ą) trapping. J. Taiwan Institute Chem. Eng. 2016, 65, 304-311.

  1. I have a question concerning the weight loss: why is the starting weight different between the materials and how often was the experiment repeated. Are these mean values?
  • Thank you for this comment. Each material had different initial weight as they were prepared by cutting them from the tube. Experiments were repeated three times and subsequently presented as the mean values what is now accordingly included in our report.
  1. A related question to the tannic acid release: in materials and methods, the authors state, that triplicate measurements were performed – does that mean, that three independent preparations of the hydrogel combinations were analysed? In my Opinion, this should be the case. The same is true for the biological experiments: at least three independent (independent material preparations, independent cell and bacteria preparations) should be analyzed to account for some biological variations.
  • Thank you for the discussion. Yes, indeed, three experiments were prepared separately by the independent hydrogels preparation. Additionally, biological studies were carried out for three independent materials, and this has been mentioned in our manuscript.
  1. For the tests on antibacterial effects, as well as for the blood and cell compatibility the hydrogel materials were sterilized by UV-irradiation, and 75% ethanol treatment, respectively. Does this treatment have effects on (i) structure/integrity of the material (e.g. does the pore size change; is tannic acid released by ethanol treatment etc.), and (ii) on the subsequent test (i.e. was the ethanol removed completely from the hydrogel, before incubating the material with the cells?)
  • Thank you for the comment. Sterilization method was not selected to assess the influence on the material properties. Respectfully to the reviewer’s suggestion, we have already determined this sterilization assay regarding other materials evaluated in our laboratory.
  1. To show an antibacterial activity of the hydrogel (in its envisioned form for example as a wound dressing), I would suggest to perform an agar diffusion test and to show the inhibition zone of the different material compositions, additionally.  
  • We agree that the agar diffusion tests are useful to show the inhibition zone of materials nevertheless, while we were conducting our study we observed that material partially dissolved during experiment. Due to this, we proposed dehydrogenase activity tests.
  1. The discussion is very basal and should be considerably extended to comprise discussions on:
  • Comparison of the results of this study to results of former studies analyzing similar materials; is the “novel” approach better, is the outcome different.

What to expect from the tannic acid-alginate hydrogel: based on the known activities of tannic acid, what would be potential modes of application.

We appreciate all comments. We propose prepared hydrogels as materials for wound treatment as wound dressing.

“It may be assumed that proposed preparation method leads to the eco-friendly and biocompatible materials obtainment mainly as wound dressings.”

“Proposed materials of alginate/tannic acid hydrogels cross-linked by calcium ions have antibacterial effect what provides great advantage comparing with pure alginate-based materials.”

Reviewer 2 Report

  1. The biomedical application for current hydrogen is mostly for wound dressing.
  2. Some abbrev is lacking the full name such as SBF, PBS.
  3. Figure 2 and 3 need standard deviation.
  4. For tannic acid release result, author could calculate loading efficiency and release rate.
  5. Table 1 need to convert to column.
  6. Tannic acid has the reduction property. For table 1, dose tannic acid would not influence the MTS result?
  7. For figure 4, the 6-day cell culture result seems not reasonable for figure 3, as tannic acid is not release after 20 hour. Could author provide the cell culture result for 12hr, 24hr or 48hr?

Author Response

  1. The biomedical application for current hydrogen is mostly for wound dressing.

  • Yes, we agree with the statement. The main biomedical application of hydrogels are wound dressings what is now added to the report:

“For such reason hydrogels based on natural polymers are mainly in the scientific research to be tested for wound healing”.

  1. Some abbrev is lacking the full name such as SBF, PBS.

  • We appreciate your comment. It is now explained as follows:

„simulated body fluids (SBF)”

“phosphate buffered saline (PBS)”

  1. Figure 2 and 3 need standard deviation.

  • Thank you for the suggestion. Figure 2 and 3 is now corrected properly.

  1. For tannic acid release result, author could calculate loading efficiency and release rate.

  • Thank you for the suggestion. We agree that such parameters as loading efficiency and release rate may be calculated. As it is presented in the paper, we calculated the released concentration of tannic acid. The loading efficiency was 100% as we added tannic acid to the sodium alginate solution and then mixture was placed in dialysis tubes for cross-linking process by calcium ions. It is now written in the paper.

  1. Table 1 need to convert to column.
  • Thank you for the suggestion. Results are now presented in column.

  1. Tannic acid has the reduction property. For table 1, dose tannic acid would not influence the MTS result?
  • During the measurement we made also blank tests to notice absorbance of tannic acid without cells during measurement and with cells. Difference was included in the results.

  1. For figure 4, the 6-day cell culture result seems not reasonable for figure 3, as tannic acid is not release after 20 hour. Could author provide the cell culture result for 12hr, 24hr or 48hr?
  • We appreciate your comments. However, to notice differences in the cell viability in contact with material we carry out experiment for 6 day. It is compatible with our previous studies. After 6 day we may truly say if the material influences on the cells viability. For the short time of cultivation we did not observe significant changes in cells viability. Moreover, as our work in lab is limited now it would be hard for us to repeat experiment. We hope that presented tests will be acceptable by Reviewer.

Round 2

Reviewer 1 Report

English language could still be improved. 

As far as I could see the discussion part was hardly changed during revision and in my opinion, a detailed discussion on these two topics is still missing: 

  • about the current knowledge on advantages or maybe disadvantages of the “novel” combination Ca2+ cross-linked tannic acid-containing alginate hydrogels as compared to other, similar alginate or other tannic acid-containing or pure hydrogels - in terms of stability, bio-compatibility, toxicity...... 

  • about the potential effects of tannic acid on wound healing (typing “tannic acid skin wound healing” into pubmed search gives more than 60 papers on that topic).

Error bars were added in Figs 2 and 3, which is appreciated, BUT why are there also horizontal error bars in the graphs?

It would make it easier for the reader, if the columns in Figure 4 were (1) grouped by S. aureus, E.coli and (2) if the pattern/ color was more distinctive.

There are now two Figures 4 in the Manuscript.

Author Response

Ref: materials-948711 (Round 2)

Title: Novel eco-friendly tannic acid-enriched hydrogels - preparation and characterization for biomedical application

Journal: Materials

Dear Editor-in-Chief and Dear Reviewers, 

We would like to thank for the comments regarding our manuscript submitted to Materials journal for review process. We would like to thank also the Editor that gave us chance to correct our manuscript. All changes made in our manuscript are marked in yellow. Please find below our answer to the comments.

Herein, we greatly appreciate all the reviewer’s comments and we have thoroughly addressed to all of them point-by-point below:

Reviewer 2

  1. English language could still be improved. 

Thank you for the suggestion. English language was checked and corrected.

  1. As far as I could see the discussion part was hardly changed during revision and in my opinion, a detailed discussion on these two topics is still missing: 
  • about the current knowledge on advantages or maybe disadvantages of the “novel” combination Ca2+ cross-linked tannic acid-containing alginate hydrogels as compared to other, similar alginate or other tannic acid-containing or pure hydrogels - in terms of stability, bio-compatibility, toxicity...... 

 We appreciate your comment. It is now written:

“Mixing sodium alginate and tannic acid with the further cross-linking process by calcium ions addition is novel and allows for preparing hydrogels by a fast and low-cost method. Obtained materials were stable at 37oC and showed significant compatibility with cells. Furthermore, prepared hydrogels had an antibacterial effect against Gram-positive and Gram-negative bacteria. It results that proposed hydrogels combine the most important features as biocompatibility, antibacterial effectiveness, and safety for the environment.”

We hope it is acceptable.

  • about the potential effects of tannic acid on wound healing (typing “tannic acid skin wound healing” into pubmed search gives more than 60 papers on that topic).

We agree with reviewer. We decided to ass sentence in the discussion part “Proposed hydrogels would protect wound against the bacterial infections and at the same time would be safe for cells, or even will enhance their proliferation”. In our opinion this is the main advantage of material what is now underlined.

  1. Error bars were added in Figs 2 and 3, which is appreciated, BUT why are there also horizontal error bars in the graphs?

We apologize for the mistake. We agree that horizontal error bars should not be in Figure 2 and 3. It is now corrected.

  1. It would make it easier for the reader, if the columns in Figure 4 were (1) grouped by S. aureus, E.coli and (2) if the pattern/ color was more distinctive.

Thank you very much for the suggestion. Figure 4 is now corrected. We hope it is acceptable in the present form.

  1. There are now two Figures 4 in the Manuscript.

We apologize for the mistake. It is now corrected.